# A Unified Data and Model-Centric Framework for Robust Facial Emotion Recognition

## Abstract

Recent Progress in Deep Learning (DL) has shown that data quality constrains the generalization as much as model design. Facial Emotion Recognition (FER) exemplifies this challenge, as widely used datasets contain mislabeled, duplicated, class imbalanced, and visually affected samples that weaken both accuracy and robustness. In this paper we proposed a data-centric approach to FER, building a systematic pipeline that improves dataset reliability before model training. The pipeline includes (i) Noisy and duplicated samples removal, (ii) landmark-guided facial refinement, and (iii) class-aware re-balanced under-presented emotions in the dataset. Following the data-centric pipeline we proposed a lightweight hybrid CNN-Transformer student model with Emotion Aware Dynamic Distillation (EADD), where knowledge is adaptively distilled from multiple teacher networks depending on their emotion-specific strengths. Despite the multi-teacher knowledge distillation student model is further optimized by adversarial training to enhance its robustness against subtle perturbations in real-world FER. Extensive experiments on FER2013 and KDEF highlights that our approach achieved state-of-the-art robustness, efficiency and trade-offs for real-time FER on Edge devices. The results demonstrate that systematic data refinement is as critical as model innovation. The source code for results reproducibility of the paper is publicly available at `https://github.com/anonymous123810/ICLR2026`.

## 1 Introduction

Facial expressions are a fundamental aspect of human communication, conveying emotions like happiness, sadness, or anger to subtle cues such as a fleeting smile or a raised eyebrow. Recognizing these expressions automatically, known as Facial Emotion Recognition (FER), which has become increasingly important in fields such as human-computer interaction, healthcare, automotive safety, and intelligent surveillance (Khan et al., 2025b). Facial expressions account for a substantial portion of non-verbal communication and the ability to accurately interpret these signals is essential for Artificial Intelligence (AI) systems that interact with human in socially aware and emotionally intelligent ways (Kaur & Kumar, 2024). Despite remarkable advancements in Deep Learning (DL), recent studies evaluated FER systems under controlled conditions, where hand-picked datasets provide clean labels, balanced classes, and consistent face regions. While real-world scenarios present far greater challenges since facial expressions differ across age, gender, cultural background, and even neurological conditions such as Parkinson's or Alzheimer's disease, which can diminish emotional cues (Munsif et al., 2024). These challenges underscores the critical importance of data-centric focused approaches, where performance improved not only by scaling the model but by addressing the underlying data quality.

Existing studies primarily focused on developing novel architectures such as Convolutional Neural Networks (CNNs) (Agung et al., 2024), transformer-based models (Xu et al., 2023), or hybrid CNN-Transformer frameworks (Tang et al., 2024). While these methods have advanced recognition performance, their effectiveness is often limited by dataset deficiencies such as mislabeled samples, duplicated or low-quality images, class imbalance, and inconsistent facial region. These problems introduces noise and bias, leading to poor generalization in practical environment, particularly deployment over resource-constraint devices. Although, recent studies have attempted to mitigate these problems through transfer learning (Zhou et al., 2024), self-supervised pretraining(Chen et al., 2020), and adversarial robustness (Nern et al., 2023). However, these methods largely adapt the

provided data rather than improving its quality which raises an important question: *Can a systematic data-centric pipeline significantly enhance FER system performance under real-world, unconstrained environment?*

To address this challenge, we propose a unified framework that integrates a data focused preprocessing pipeline, a lightweight hybrid CNN–Transformer (HyFER) architecture, and multi-phase training strategy designed to improve robustness, generalization, and real-time applicability. The preprocessing pipeline systematically enhances the quality of the KDEF (Calvo & Lundqvist, 2008) and FER2013 (Courville et al., 2013) datasets by extracting facial regions , removing mislabeled and duplicate samples, and applying class-specific upsampling to mitigate the bias in model predictions caused by underrepresented emotion classes. Building upon these refined datasets, we train a lightweight HyFER student model, explicitly designed to capture both fine-grained local facial textures and global contextual dependencies. Moreover, the framework employs a dual-phase optimization strategy, combining multi-teacher knowledge distillation with post-distillation adversarial training, to ensure stable FER under challenging real-world conditions such as occlusion, noise, and varying illumination.

## 2 RELATED WORK

Automatic identification of facial emotions has attracted significant attention due to its vital role in transferring human emotions to machine perception; yet, FER systems are facing challenges including variability in facial expressions, environmental factors, and constraints dataset. However, several studies on FER largely focused on handcrafted features and conventional Machine Learning (ML) approaches. Descriptor such as Histogram of Oriented Gradient (HOG) (Carcagnì et al., 2015), Local Binary Patterns (LBP) (Shan et al., 2009), Scale-invariant Feature Transformer (SIFT) (Soyel & Demirel, 2011), Speed-up Robust Features (SURF) (Rao et al., 2015), and Gabor filters (Lyons et al., 2020) were frequently used to capture local facial textures and directional changes. These feature extractors were often integrated with classifiers like Support Vector Machines (SVM), and occasionally with the Facial Action Coding System (FACS) (Pantic & Rothkrantz, 2004) to translate expression into action units. Despite their effectiveness in controlled settings, conventional approaches weren't robust against real-world variability including posture, lighting, and occlusion.

The advancements in DL have revolutionized FER by allowing ML models to learn complex features from unprocessed facial shots, rather than manually selected feature (Huang et al., 2017; Szegedy et al., 2016; 2017). This revolution in DL began with the development of Convolutional Neural Networks (CNNs) based models such as VGGNet (Simonyan & Zisserman, 2014) which extracts both low-level and high-level features more accurately, but their utility in real-time applications was restricted due to their high computational costs. Subsequent studies investigated lightweight CNNs (Huo et al., 2023; Saurav et al., 2022) and dual-stream pipelines (Sarvakar et al., 2023) to minimize complexity while maintaining discriminative capacity. Other studies have used temporal modeling with RNNs and Transformer (Ullah et al., 2022; Liang et al., 2020a), as well as multimodal fusion approaches proposed in (Sun et al., 2019) to capture dynamic expressions across video sequences. Moreover, Ensemble-based techniques (Wadhawan & Gandhi, 2022; Moung et al., 2022; Khan et al., 2025b) further improved performance by combining complementary feature extractors and attention mechanisms.

Considering these advancements, FER models are still highly sensitive to data quality. Noisy labels, class imbalance, and loosely cropped samples in widely utilized benchmarks such as FER2013 and KDEF propagate bias into learnt models (Nguyen et al., 2022). Recent data-centric method have employed various methods to enhance datasets diversity by utilizing targeted class transformation (Zhu et al., 2022). While data quality enhancement in terms of label correction, and duplicated sample removal for FER remains unexplored.

Beyond data quality enhancement, model-centric techniques aimed to develop an efficient and robust model. In context to develop an optimized computationally efficient model Knowledge Distillation (KD) has emerged as a more common approach to distill the rich knowledge from computationally expensive model to less computationally expensive model (Hinton et al., 2015a). Moreover, to improve the generalizability of the model for real-word unconstrained situation under perturbations and noisy conditions adversarial training is proposed by (Zheng et al., 2020).

Conclusively, prior studies outline the dual challenges of data-centric and model-centric shortcomings in FER. Although deep architectures, and ensemble strategies have advanced the field, limited attention has been given to unified frameworks that simultaneously address dataset quality, lightweight architecture design, and robust training under real-world conditions. Our work addresses these issues by proposing a multi-phase pipeline that includes systematic preprocessing, a HyFER hybrid model architecture, and a dual-phase KD-adversarial optimization approach.

## 3 PROPOSED METHODOLOGY

In this section, we present the proposed data-centric and model-centric pipeline design to develop a robust and computationally efficient FER system, optimized for real-time deployment on embedded devices. The framework incorporates a data-centric preprocessing pipeline design to construct a clean, balanced, and high-quality emotion corpus. This is followed by a dual-phase optimization strategy that incorporates Multi-Teacher Knowledge Distillation (MTKD) with post-distillation adversarial training. The high level overview of the unified framework is depicted in Figure 1. Moreover, the detailed explanation of each component in the proposed framework is elaborated in the subsequent subsection.

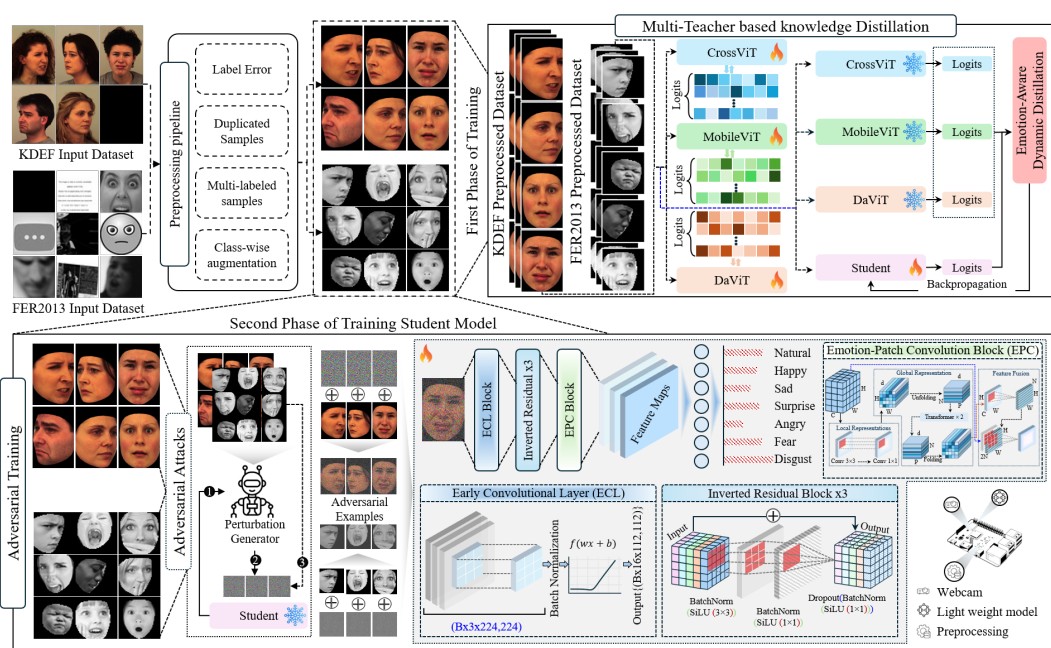

Figure 1: The high level overview of the unified proposed framework.

### 3.1 DATA-CENTRIC PREPROCESSING PIPELINE

In This study we utilize two publicly available facial expression datasets named as FER2013 (Zahara et al., 2020) and KDEF (Eng et al., 2019), to train and evaluate the proposed lightweight HyFER student model. Despite the benchmark popularity, these datasets exhibit several data quality and distributional challenges, including high intra class variance due to noisy samples, annotation inconsistencies leading to label error, redundant samples resulting in data duplication, ambiguous label associations where multiple emotions are assigned to a single instance, multilabel samples, and significant class imbalance that skews the learning process. To mitigate these challenges and enhance the quality of the input data, this study employed a comprehensive data preprocessing pipeline designed to transform the raw data into a more informative and structured format facilitating optimal learning and improved performance across multiple evaluation indicators. The detailed preprocessing steps are elaborated in the following sections.

### 3.1.1 IMAGE QUALITY ENHANCEMENT

Image quality enhancement pipeline employed in this study aims to mitigate the visual inconsistencies from FER2013 and KDEF benchmarks. To ensure reliable input for the DL models' optimization irrelevant, noisy, duplicated, and multilabel samples were removed by visually inspecting the samples to enhance dataset consistency. However, duplicated instances were detected through both inter-class and intra-class cosine similarity assessments, followed by manual visual inspection to verify redundancy while preserving dataset integrity. Although the FER2013 dataset exhibited numerous quality issues, resulting in an overall error rate of 6.64%, the KDEF dataset was comparatively cleaner, with a significantly lower error rate of 0.0624% and only four samples lacking recognizable facial emotions, which were excluded during preprocessing, as summarized in Table 1. In addition to removing problematic and low-quality samples, facial region extraction was employed in the image quality enhancement pipeline using pretrained MediaPipe Face Mesh Detector (Lugaresi et al., 2019) to further refine the input data by generating facial mask from the predicted landmarks, isolating the facial region while discarding the background. This step enabled the pipeline to isolate and focus on the most informative regions, eliminating background noise and non-facial regions that could interfere with learning. The proposed image quality enhancement pipeline enhances facial image quality by removing noisy samples, resolves label inconsistencies, and isolating high-fidelity facial features, thereby enhancing data reliability, stabilizing the training process, and improved model generalization. The discarded low-quality samples from FER2013 and KDEF benchmarks are illustrated in Figure 2.

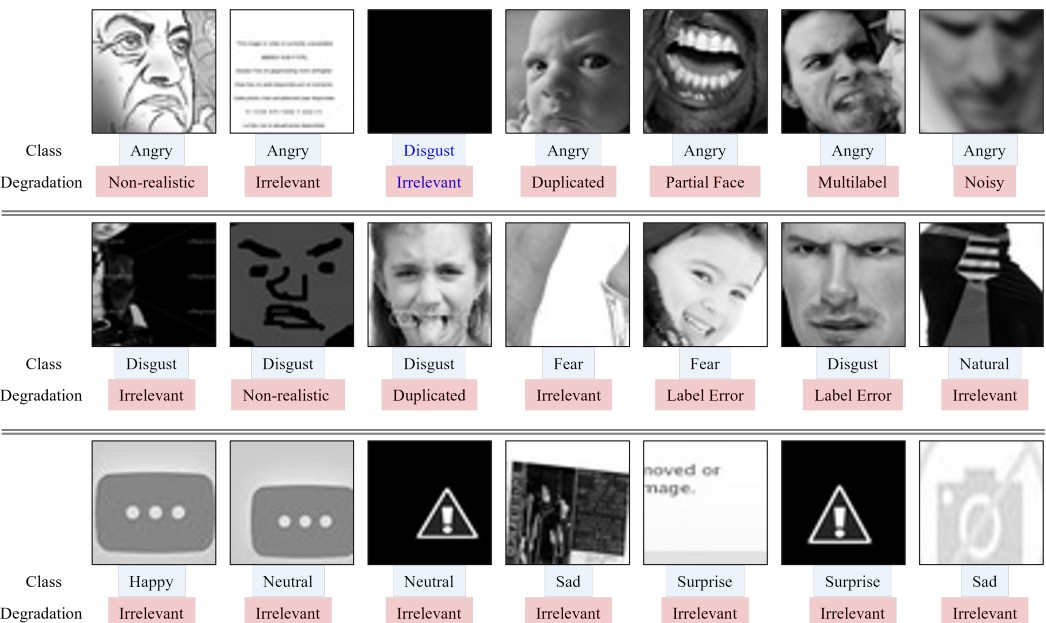

Figure 2: Low-quality samples from FER2013 and KDEF benchmarks.

### 3.1.2 CLASS-SPECIFIC DATA AUGMENTATION

This subsection presents the class-targeted data augmentation approach which allows to address class imbalance within the benchmarks, a condition that typically leads to model bias toward majority classes and underperform on minority classes due to insufficient representation of minority classes while preserving the integrity of majority of class samples. This approach aims to improve model generalization and robustness by ensuring equitable learning across all classes (Yar et al., 2025). To achieve this, augmentation techniques were applied in a class-aware manner based on distributional characteristics of each class, thereby reducing bias and improving the model's ability to learn more discriminative features. The geometric transformation which have been applied including horizontal flipping, vertical flipping, controlled rotation ±10°, translation up to 5%, and

Table 1: Statistical analysis of the FER2013 and KDEF datasets including label error, error ratio, original sample counts, and augmented sample counts

| Class | Duplicated Samples (FER2013) | Multilable Samples (FER2013) | Label Error | | Error Ratio (%) | | Org. Sample Count | | Augmented Sample Count | |
|---|---|---|---|---|---|---|---|---|---|---|
| | | | FER2013 | KDEF | FER2013 | KDEF | FER2013 | KDEF | FER2013 | KDEF |
| Angry | 126 | 10 | 473 | N/A | 12.29 | N/A | 4,953 | 840 | 8,341 | 961 |
| Disgust | 27 | N/A | 27 | 2 | 9.872 | 0.2178 | 547 | 918 | 8,226 | 956 |
| Fear | 145 | 24 | 193 | N/A | 7.068 | N/A | 5,121 | 762 | 8,512 | 970 |
| Happy | 75 | N/A | 232 | N/A | 3.415 | N/A | 8,989 | 858 | N/A | 961 |
| Neutral | 59 | 4 | 40 | 2 | 1.661 | 0.2192 | 6,198 | 912 | 8,425 | 953 |
| Sad | 63 | 9 | 55 | N/A | 2.089 | N/A | 6,077 | 975 | 8,627 | N/A |
| Surprise | 350 | 15 | 39 | N/A | 10.09 | N/A | 4,002 | 603 | 8,517 | 935 |

scaling within the range of $0.9\times$ to $1.1\times$. In addition, color and texture transformation were employed to improve the model robustness to variations in illumination and sensor noise, particularly for minority classes characterized by insufficient illumination variability. The class-specific sample count of class-aware data augmentation preprocessing pipeline is shown in Table 1.

### 3.2 DUAL-PHASE OPTIMIZATION

The dual-phase optimization framework proposed for FER integrates MTKD with Post-distillation adversarial training aiming to optimize the proposed computationally efficient HyFER model which is detailed in the subsequent sections.

#### 3.2.1 MULTI TEACHER COLLABORATIVE LEARNING

Collaborative learning in KD intent to distill comprehensive feature representations from multiple teacher networks into lightweight student model. The complementary strengths of multiple fine-tuned teacher networks facilitate multi-faceted emotional features representations such as global facial structures, fine-grained expression details, and contextual cues collaborating to provide a rich knowledge base for the student model (Hinton et al., 2015b). In MTKD collaborative framework the knowledge is transferred into the lighter student model through a novel Emotion-Aware Dynamic Distillation (EADD) framework which dynamically prioritizes teacher contributions based on their expertise in specific emotion (e.g., happy, sad, angry). The EADD optimizes the HyFER model through a composite loss function that integrates a standard cross-entropy loss with a dynamic, emotion-specific distillation loss, as computed by:

$$L_{\text{EADD}} = \alpha L_{\text{CE}}(y, \hat{y}_s) + \sum_{e=1}^{E} \sum_{i=1}^{N} \omega_{i,e}(t) \, L_{\text{Distill}}\left(z_s, z_i^T, T_e\right) \tag{1}$$

Where $L_{\text{CE}}$ denotes cross-entropy loss, measuring the difference between the ground truth $y$ and student prediction $\hat{y}_s$ mathmathically formulated in Eq. 2. The term $L_{\text{Distill}}$ represents the KD loss between the student and $i^{\text{th}}$ teacher model over emotional expression $e$ computed using student and teacher networks logits $z_s$ and $z_{T_{i,e}}$, respectively, computed as Eq. 3.

$$L_{CE}(y, \hat{y}_s) = -\sum_{c=1}^{C} y_c \log\left(\hat{y}_{s,c}\right) \tag{2}$$

$$L_{\text{Distill}}(z_s, zT_i, T_e) = T_e^2 \cdot \text{KL}\left(\text{Softmax}\left(\frac{zT_{i,e}}{T_e}\right), \text{Softmax}\left(\frac{z_s}{T_e}\right)\right) \tag{3}$$

The distillation loss is computed using the Kullback-Leibler (KL) divergence (Hershey & Olsen, 2007) mathematically computed by Eq. 4, which quantifies the discrepancy between the teacher's softened class probability $PT_{i,e,c}$, and the student's softened probabilities $P_{s,c}$ over emotion $c$.

$$\mathrm{KL}\big(PT_{i,e} \parallel P_s\big) = \sum_{c=1}^{C} PT_{i,e,c} \, \log\left(\frac{PT_{i,e,c}}{P_{s,c}}\right) \tag{4}$$

Moreover, the dynamic weighting factor in proposed composite loss function for $i^{\mathrm{th}}$ teacher over emotion $e$ at training step $t$ for $N$ number of teacher over temperature $\tau_e$ is denoted by $\omega_{i,e}(t)$ formulated in Eq. 5 measured based on the teacher's validation performance to prioritize emotion specific teacher network.

$$\omega_{i,e}(t) = \frac{\exp\left(\mathrm{Acc}_{i,e}(t)/\tau_e\right)}{\sum_{j=1}^{N} \exp\left(\mathrm{Acc}_{j,e}(t)/\tau_e\right)} \tag{5}$$

In addition to the efficiency of EADD composite loss, the framework is driven by the strategic selection of multiple teacher models. Given the multifaceted nature of emotional expressions, which includes global facial structures, fine-grained expression details, and contextual cues, a single teacher network is insufficient to capture the full spectrum of facial emotional nuances. To address this, the proposed methodology employed an ensemble of three teacher networks including DaViT (Ding et al., 2022a), CrossViT (Chen et al., 2021a), and MobileViT (Mehta & Rastegari, 2021a) selected based on the extensive experiments conducted over benchmarks.

### 3.2.2 POST-DISTILLATION ADVERSARIAL TRAINING

Building upon the EADD composite loss function and MLKD framework which effectively transfers emotional knowledge from DaViT, CrossViT, and MobileViT to the lightweight HyFER model this section focuses to present the model's resilience for real-world environment. Although EADD equips HyFER with rich, multi-scale emotional representation optimized for resource-constrained devices, real-world FER systems necessitate a high degree of robustness under a wide range of challenging conditions, including image noise, illumination variability, and imperceptible adversarial perturbations. To mitigate these challenges in real-world FER systems, a post-distillation adversarial training phase is incorporated to enhance the robustness and generalization capability of the lightweight model without compromising its computational efficiency. This phase employed several white-box adversarial attack algorithms during training to further refine the optimized HyFER model, including Projected Gradient Descent (PGD) (Ren et al., 2020), Fast Gradient Sign Method (FGSM) (Yinusa & Faezipour, 2025), and DeepFool (Moosavi-Dezfooli et al., 2016). PGD white-box attack is an iterative attack which generates adversarial examples by iteratively perturbing the input image in the direction to maximize the model's loss over predefined number of steps, as formulated below.

$$x^{(t+1)} = \prod_{B \in (x)} \left( x^{(t)} + \alpha \cdot \mathrm{sign}\left(\nabla_x L\big(f_\theta(x^{(t)}), y\big)\right) \right) \tag{6}$$

Here, $x^{(t)}$ denotes the adversarial example at iteration $t$, while $x^{(t+1)}$ represents the updated adversarial input over computed cross-entropy loss $L$ between the model prediction $f_\theta(x^{(t)})$ and ground truth label $y$.

In contrast to the iterative approach of PGD, the FGSM provides a computationally efficient, single step approach for generating adversarial examples, as defined below.

$$x_{\mathrm{adv}} = x + \epsilon \cdot \mathrm{sign}\left(\nabla_x J(\theta, x, y)\right) \tag{7}$$

Where $x$ denotes the input, and $\epsilon$ is a small scalar that defines the magnitude of the perturbation. The expression $\nabla_x J(\theta, x, y)$ represents the gradient of the cross-entropy loss with respect to the input $x$.

Furthermore, DeepFool white-box attack is designed to compute the minimal perturbation required to alter decision boundaries of a classifier. Unlike PGD and FGSM that rely on iterative and predefined magnitude, DeepFool approach formulates the attack as an optimization problem that iteratively estimates the classifier's decision boundaries and determines the smallest possible perturbation required to change the model's prediction, as detailed in (Moosavi-Dezfooli et al., 2016).

### 3.3 Lightweight Hybrid Student Model

The architectural representation of the student model is essential to ensure real-time accurate FER over resource-constrained devices. The student model is developed as compact hybrid network that effectively combines convolution operation with transformer-based components to balance computational efficiency while improve performance as diagrammatically represented in Figure 1. The architectural design of it is inspired from MobileViT (Mehta & Rastegari, 2021a) which begins with Early Convolutional Layer (ECL) consisting a three-by-three kernal with stride two followed by batch normalization and SiLU activation function. Following the ECL projection block the network is structured into three sequential inverted residual blocks which are configured by one-by-one, three-by-three and one-by-one constitutional operations. This block ensures the backbone to be parameter efficent and expressive. Following the third inverted residual block, the architecture integrates the proposed Emotion-Patch Convolution (EPC) block, which is adapted from MobileViT architectural design. The feature maps projected by the EPC block are subsequently passed trough a projection layer, which consists of a one-by-one convolution that increases the channel dimension, followed by batch normalization, a SiLU activation, and dropout. This projection enriches the representational capacity of the network while preparing the features for the output layer. The output classifier head of the model is configured by global average pooling operation, and regularized with dropout before being passed into a fully connected layer that maps the features to seven output categories, corresponding to the emotion classes under consideration.

## 4 Results And Discussion

This section details the implementation details, experimental setup, dataset, and evaluation metrics for the proposed framework. The MTKD, subsequent post-adversarial optimization and model development were implemented in PyTorch 2.6, using Adam optimizer. The dual-phase optimization pipeline was trained for 100 epochs with a batch size of 16 and input resolution of $224 \times 224 \times 3$. All experiments including Fine-tuning of the multi teacher, training and evaluation of the HyFER lightweight student model were carried out on a computing system configured with an NVIDA GeForce RTX 3090 GPU (12 GB VRAM) and 128 GB of system RAM.

**Evaluation Metric.** To assess the effectiveness of the dual-phase optimization framework against state-of-the-art method we used several indicators, including accuracy, precision, recall, and F1-score following (Khan et al., 2025a).

**Dataset.** The selected teacher networks and student model is evaluated over KDEF (Calvo & Lundqvist, 2008) and FER2013 (Courville et al., 2013) dataset. FER2013 comprises 35,887 grayscale images of of size $48 \times 48$ resolution across seven emotion classes[1], collected via the Google image search API. Similarly, KDEF dataset contains 4,900 samples representing seven basic emotions[1], collected from 70 participants in a controlled laboratory environment by Karolinska Institute. The class-wise statistical analysis of these benchmarks are presented in Table 1.

### 4.1 Performance Evaluation

This section presents performance analysis of the HyFER student model and baseline methods, optimized through proposed dual-phase optimization paradigm under K-fold cross-validation setup to ensure robust and reliable evaluation, as reported in Table 2. The table demonstrated that methods such as EA-Net (Khan et al., 2025b) and GA (Nida et al., 2024) achieved higher performance as compared to other baseline methods over KDEF dataset containing controlled laboratory images. On the other hand, FER2013 dataset, which presents more challenging samples varying illumination, and occlusion, methods such as CBiLSTM (Liang et al., 2020b) and DBN (Vedantham & Reddy, 2020) exhibits significant drops in performance especially in precision and F1-score, highlighting their limited generalization ability. Moreover other baselines such as PIDViT (Huang & Tsai, 2022) and EA-Net (Khan et al., 2025b) demonstrate improved robustness but still fall short in comparison to our proposed HyFER student model. In conclusion, these results indicates that the carefully designed architecture of the lightweight student model, coupled with dual-phase optimization, ensures consistent and robust performance in both controlled and unconstrained real-world settings.

---

[1]The seven facial emotion include angry, disgust, fear, happy, sad, surprise, and neutral.

Table 2: Baseline and proposed method evaluation under the dual-phase (multi-teacher KD + Post-KD adversarial training) optimization framework with 5-fold cross-validation (K=5). Results are reported as mean $\pm$ stander deviation for evaluation metrics including accuracy, precision, recall, and F1-score over KDED and FER2013 dataset. Our proposed method results is highlighted in bold.

| Method | KDEF | | | | FER2013 | | | |
|---|---|---|---|---|---|---|---|---|
| | Accuracy | Precision | Recall | F1-score | Accuracy | Precision | Recall | F1-score |
| FMA+SVM (Solis-Arrazola et al., 2024) | $0.725 \pm 0.020$ | $0.713 \pm 0.016$ | $0.724 \pm 0.019$ | $0.718 \pm 0.017$ | $0.589 \pm 0.016$ | $0.576 \pm 0.019$ | $0.583 \pm 0.021$ | $0.564 \pm 0.015$ |
| FMA+MLP (Solis-Arrazola et al., 2024) | $0.726 \pm 0.050$ | $0.704 \pm 0.090$ | $0.716 \pm 0.046$ | $0.692 \pm 0.062$ | $0.582 \pm 0.065$ | $0.568 \pm 0.037$ | $0.594 \pm 0.024$ | $0.586 \pm 0.052$ |
| FMA+LD (Solis-Arrazola et al., 2024) | $0.764 \pm 0.014$ | $0.752 \pm 0.032$ | $0.763 \pm 0.025$ | $0.754 \pm 0.021$ | $0.615 \pm 0.045$ | $0.605 \pm 0.024$ | $0.613 \pm 0.015$ | $0.600 \pm 0.026$ |
| DBN (Vedantham & Reddy, 2020) | $0.885 \pm 0.013$ | $0.872 \pm 0.017$ | $0.862 \pm 0.015$ | $0.865 \pm 0.018$ | $0.647 \pm 0.020$ | $0.615 \pm 0.020$ | $0.639 \pm 0.019$ | $0.627 \pm 0.017$ |
| CBiLSTM (Liang et al., 2020b) | $0.932 \pm 0.036$ | $0.916 \pm 0.046$ | $0.925 \pm 0.062$ | $0.915 \pm 0.022$ | $0.582 \pm 0.029$ | $0.556 \pm 0.043$ | $0.572 \pm 0.052$ | $0.565 \pm 0.047$ |
| Joint-Attention (Ghaleb et al., 2023) | $0.963 \pm 0.026$ | $0.926 \pm 0.046$ | $0.954 \pm 0.036$ | $0.946 \pm 0.043$ | $0.743 \pm 0.036$ | $0.724 \pm 0.025$ | $0.726 \pm 0.036$ | $0.736 \pm 0.062$ |
| H-attention (Tao & Duan, 2024) | $0.972 \pm 0.047$ | $0.956 \pm 0.066$ | $0.966 \pm 0.046$ | $0.975 \pm 0.035$ | $0.746 \pm 0.033$ | $0.733 \pm 0.075$ | $0.740 \pm 0.105$ | $0.733 \pm 0.095$ |
| PIDViT (Huang & Tsai, 2022) | $0.973 \pm 0.036$ | $0.962 \pm 0.054$ | $0.975 \pm 0.067$ | $0.975 \pm 0.043$ | $0.763 \pm 0.033$ | $0.754 \pm 0.073$ | $0.738 \pm 0.033$ | $0.748 \pm 0.026$ |
| MTAC (Zhang et al., 2023) | $0.975 \pm 0.064$ | $0.965 \pm 0.073$ | $0.973 \pm 0.073$ | $0.973 \pm 0.057$ | $0.726 \pm 0.048$ | $0.716 \pm 0.047$ | $0.726 \pm 0.033$ | $0.705 \pm 0.021$ |
| Hit-mst (Xia & Jiang, 2023) | $0.985 \pm 0.036$ | $0.975 \pm 0.046$ | $0.973 \pm 0.074$ | $0.983 \pm 0.043$ | $0.773 \pm 0.064$ | $0.764 \pm 0.054$ | $0.752 \pm 0.043$ | $0.743 \pm 0.047$ |
| GA (Nida et al., 2024) | $0.985 \pm 0.045$ | $0.975 \pm 0.024$ | $0.967 \pm 0.032$ | $0.985 \pm 0.043$ | $0.775 \pm 0.043$ | $0.763 \pm 0.053$ | $0.765 \pm 0.073$ | $0.769 \pm 0.063$ |
| EA-Net (Khan et al., 2025b) | $0.992 \pm 0.053$ | $0.996 \pm 0.033$ | $0.982 \pm 0.062$ | $0.991 \pm 0.026$ | $0.760 \pm 0.063$ | $0.770 \pm 0.074$ | $0.790 \pm 0.036$ | $0.780 \pm 0.062$ |
| **Proposed (ours)** | $\mathbf{0.996 \pm 0.015}$ | $\mathbf{0.997 \pm 0.026}$ | $\mathbf{0.987 \pm 0.013}$ | $\mathbf{0.982 \pm 0.019}$ | $\mathbf{0.794 \pm 0.024}$ | $\mathbf{0.786 \pm 0.015}$ | $\mathbf{0.776 \pm 0.036}$ | $\mathbf{0.785 \pm 0.022}$ |

## 4.2 ABLATION STUDY

In this section, we investigate the multi-teacher network selection, dual-phase optimization framework and the HyFER student model computational cost in terms of GFLOPs, number of parameter count and model size. Moreover, evaluation assessments of the teacher network selection and HyFER were conducted under K-fold cross-validation.

**Multi-Teacher Network Selection.** Teacher networks for MTKD were selected through extensive experiments evaluation on the transformer-based models over benchmarks, as summarizes in Table 3. The teacher network are selected based on the recognition performance, and computational cost. The selected networks include CrossViT-18 (Chen et al., 2021b), DaViT (Ding et al., 2022b), and MobileViT-S (Mehta & Rastegari, 2021b) demonstrate high recognition performance while maintaining minimal computational overhead. The selected multi-teacher ensures that the HyFER model effectively inherits rich multi-faced knowledge over MTKD optimization.

Table 3: Teacher models performance evaluation across FER 2013 and KDEF benchmarks. By including the key metrices such as testing accuracy, loss values, number of parameters, computational complexity (GFLOPs) and model size in megabytes (MB). Selected teacher networks are highlighted in bold.

| Teacher Models | FER2013 | | KDEF | | Params (M) | GFLOPs | Size (MB) |
|---|---|---|---|---|---|---|---|
| | Acc (%) | Loss | Acc (%) | Loss | | | |
| ConViT-S (d'Ascoli et al., 2021) | $67.99 \pm 0.42$ | $0.844 \pm 0.012$ | $92.52 \pm 0.28$ | $0.308 \pm 0.007$ | 27.35 | 5.35 | 104.32 |
| **CrossViT-18 (Chen et al., 2021b)** | $\mathbf{78.86 \pm 0.31}$ | $\mathbf{0.352 \pm 0.008}$ | $\mathbf{99.54 \pm 0.14}$ | $\mathbf{0.086 \pm 0.004}$ | **42.60** | **8.21** | **162.51** |
| FastViT-SA24 (Vasu et al., 2023) | $69.51 \pm 0.38$ | $0.799 \pm 0.010$ | $94.22 \pm 0.33$ | $0.279 \pm 0.006$ | 20.54 | 2.89 | 78.34 |
| EfficientViT-M2 (Liu et al., 2023) | $69.32 \pm 0.35$ | $0.775 \pm 0.011$ | $93.20 \pm 0.31$ | $0.316 \pm 0.008$ | 3.96 | 0.20 | 15.12 |
| **DaViT-B (Ding et al., 2022b)** | $\mathbf{77.62 \pm 0.29}$ | $\mathbf{0.562 \pm 0.009}$ | $\mathbf{97.92 \pm 0.26}$ | $\mathbf{0.108 \pm 0.005}$ | **86.94** | **15.22** | **331.64** |
| LeViT-192 (Graham et al., 2021) | $67.58 \pm 0.45$ | $0.841 \pm 0.012$ | $87.07 \pm 0.39$ | $0.417 \pm 0.009$ | 10.18 | 0.61 | 38.84 |
| MaxViT-S (Tu et al., 2022) | $68.82 \pm 0.41$ | $0.788 \pm 0.012$ | $89.80 \pm 0.34$ | $0.407 \pm 0.010$ | 67.96 | 11.27 | 260.03 |
| **MobileViT-S (Mehta & Rastegari, 2021b)** | $\mathbf{77.49 \pm 0.32}$ | $\mathbf{0.570 \pm 0.010}$ | $\mathbf{98.60 \pm 0.27}$ | $\mathbf{0.096 \pm 0.004}$ | **4.94** | **1.42** | **3.64** |

**Multi-Teacher Guided Student Optimization.** To distill rich knowledge from multiple teacher networks into the student model, we evaluate both teacher and student performance before and after the data-centric preprocessing pipeline, as detailed in Table 4. Additionally, the knowledge is progressively transferred from teacher to the student model from a single teacher to three teachers networks. The reported results demonstrate that MTKD significantly improves student model performance across both datasets, highlighting that multi-teacher KD not only enhances the discriminative capability of the lightweight student model but also stabilizes training, leading to consistent improvements across both benchmark datasets.

**Student Model Post-Distillation Adversarial Optimization.** The HyFER student model, initially optimized via the multi-teacher networks, further refinement of HyFER models for real-world applicability is enhanced through diverse adversarial perturbation methods, including FGSM, PGD

Table 4: Performance evaluation of the selected teacher networks and HyFER MTKD Before Pre-processing (BP) and After preprocessing (AP) over FER2013 and KDEF benchmarks under 5-fold cross-validation (K=5).

| Teacher Models | Acc % (BP) | | Loss (BP) | | Acc % (AP) | | Loss (AP) | |
|---|---|---|---|---|---|---|---|---|
| | FER2013 | KDEF | FER2013 | KDEF | FER2013 | KDEF | FER2013 | KDEF |
| CrossViT-18 (T1) | $69.61 \pm 0.42$ | $93.48 \pm 0.36$ | $1.707 \pm 0.051$ | $0.669 \pm 0.027$ | $78.86 \pm 0.31$ | $99.54 \pm 0.14$ | $0.352 \pm 0.008$ | $0.086 \pm 0.004$ |
| MobileViT-S (T2) | $68.98 \pm 0.37$ | $92.27 \pm 0.41$ | $1.736 \pm 0.046$ | $0.826 \pm 0.033$ | $77.49 \pm 0.32$ | $98.60 \pm 0.27$ | $0.570 \pm 0.010$ | $0.096 \pm 0.004$ |
| DaViT-B (T3) | $77.62 \pm 0.29$ | $91.93 \pm 0.42$ | $1.713 \pm 0.043$ | $0.850 \pm 0.029$ | $77.62 \pm 0.29$ | $97.92 \pm 0.26$ | $0.562 \pm 0.009$ | $0.108 \pm 0.005$ |
| HyFER (No KD) | $51.52 \pm 0.61$ | $82.99 \pm 0.57$ | $3.503 \pm 0.092$ | $1.944 \pm 0.053$ | $61.96 \pm 0.47$ | $89.07 \pm 0.34$ | $2.026 \pm 0.081$ | $0.995 \pm 0.024$ |
| HyFER (KD: T1) | $64.01 \pm 0.48$ | $92.52 \pm 0.36$ | $2.095 \pm 0.067$ | $0.804 \pm 0.022$ | $76.19 \pm 0.42$ | $98.92 \pm 0.18$ | $0.825 \pm 0.027$ | $0.190 \pm 0.010$ |
| HyFER (KD: T1+T2) | $66.14 \pm 0.44$ | $93.26 \pm 0.29$ | $1.948 \pm 0.059$ | $0.340 \pm 0.012$ | $77.92 \pm 0.38$ | $99.03 \pm 0.10$ | $0.632 \pm 0.021$ | $0.101 \pm 0.005$ |
| **HyFER (KD: T1+T2+T3)** | $\mathbf{70.25 \pm 0.28}$ | $\mathbf{94.92 \pm 0.31}$ | $\mathbf{1.440 \pm 0.067}$ | $\mathbf{0.148 \pm 0.012}$ | $\mathbf{79.39 \pm 0.25}$ | $\mathbf{99.50 \pm 0.15}$ | $\mathbf{0.283 \pm 0.009}$ | $\mathbf{0.056 \pm 0.003}$ |

and DeepFool. Table 5 reports HyFER model performance across benchmarks under K-fold cross-validation, including accuracy, loss, precision, recall and F1-score. Evolution under white-box attacks reveals that the student model consistently maintains strong performance across all metrics, achieving the highest robustness over FGMS gradient-base single-step attack perturbed samples. In contrast, PGD and DeepFool attacks adversarial examples leads to slightly lower performance, reflecting the increased difficulty posed by these method in generating strong perturbed samples.

Table 5: Performance evaluation of the optimized student model trained with knowledge distillation under various adversarial perturbations across benchmarks, reporting 5-fold cross-validation (K=5) over accuracy, loss, Precision, Recall and F1-score indicators. The best performance achieved by HyFER model is highlighted in bold.

| Dataset | Perturbation | Accuracy | Loss | Precision | Recall | F1-Score |
|---|---|---|---|---|---|---|
| | **FGSM** | $\mathbf{78.83 \pm 0.31}$ | $\mathbf{0.289 \pm 0.008}$ | $\mathbf{78.22 \pm 0.027}$ | $\mathbf{78.80 \pm 0.023}$ | $\mathbf{77.52 \pm 0.019}$ |
| FER2013 | PGD | $78.41 \pm 0.28$ | $0.295 \pm 0.009$ | $78.35 \pm 0.025$ | $77.10 \pm 0.021$ | $76.58 \pm 0.022$ |
| | DeepFool | $78.56 \pm 0.30$ | $0.291 \pm 0.007$ | $76.96 \pm 0.020$ | $77.57 \pm 0.024$ | $78.42 \pm 0.018$ |
| | **FGSM** | $\mathbf{99.43 \pm 0.12}$ | $\mathbf{0.058 \pm 0.003}$ | $\mathbf{98.92 \pm 0.009}$ | $\mathbf{99.38 \pm 0.010}$ | $\mathbf{99.01 \pm 0.007}$ |
| KDEF | PGD | $99.34 \pm 0.14$ | $0.060 \pm 0.004$ | $99.27 \pm 0.011$ | $99.18 \pm 0.012$ | $98.19 \pm 0.010$ |
| | DeepFool | $99.39 \pm 0.13$ | $0.059 \pm 0.003$ | $99.73 \pm 0.008$ | $98.23 \pm 0.009$ | $99.25 \pm 0.008$ |

# 5 CLOSING REMARKS

In this work, we presented a unified framework focusing on data quality and model design and optimization for robust real-world FER system. The data quality enhancement pipeline aimed to address critical limitations such as noisy/duplicated sample removal, landmark guided facial refinement, and class-aware rebalancing in widely used FER2013 and KDEF benchmarks. The data refinement pipeline is followed by the model-centric phase introduces HyFER, a lightweight hybrid CNN-Transformer model optimized through a dual-phase optimization strategy that combines multi-teacher knowledge distillation with post-KD adversarial training. This unified framework improved the generalization capability of the lightweight HyFER model, making it suitable for real-world FER systems. In the future, we aim to scale up this framework tackle larger-scale FER benchmarks, incorporating multimodal emotion cues, and dig into even more efficient optimization techniques.

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
