# OpenReview forum: "A Unified Data and Model-Centric Framework for Robust Facial Emotion Recognition"
_ICLR.cc/2026/Conference — ICLR 2026 Conference Withdrawn Submission_

### Official Review · Reviewer_Y3qM · 2025-10-25

**Soundness:** 1
**Presentation:** 2
**Contribution:** 1
**Rating:** 0
**Confidence:** 5

**Summary:**

This paper trains a lightweight hybrid CNN-Transformer model on two datasets: FER2013 and KDEF. During the training process, existing training techniques—multi-teacher distillation and adversarial training—are employed to enhance the robustness of representations.
The research problem addressed in this paper is based on early datasets in the face expression recognition, ignoring recent research advances.
In terms of methodology, the paper lacks new technical contributions.

**Strengths:**

- The paper is relatively clearly written.
- The experimental setup effectively illustrates the rationale for teacher selection.

**Weaknesses:**

Rather than developing and exploring new technologies, this paper is more like a research exercise; its technical contributions fall far short of the standards required by ICLR.
1. The abstract claims the paper focuses primarily on a "data-centric approach". However, the data preprocessing techniques proposed are, in fact, common data augmentation methods in deep learning-based facial expression recognition.
2. Regarding research motivation, the paper identifies issues such as "mislabeled, duplicated, and class-imbalanced data" in the FER2013 and KDEF datasets. Yet FER2013 and KDEF were released in 2008 and 2013 respectively, and these data issues are well-recognized consensus in the facial expression recognition field. It is precisely based on this consensus that the community has developed higher-quality datasets like FERPlus, AffectNet, and RAF-DB.
3. If the authors intend to study the problem of noisy labels, they should include validation experiments on datasets such as AffectNet and RAF-DB, and compare their results with the State-of-the-Art performance on these datasets.
4. The model's training methods and structural design are merely simple combinations of existing approaches, and there is a lack of justification for why such training strategies or structural designs were adopted.

**Questions:**

I suggest that the authors conduct a systematic review of the current developments in the facial expression recognition field, particularly the FER methods published in top-tier conferences. Meanwhile, they should learn the fundamental methodologies of academic research:
1. The proposed research problem must be supported by a sufficient literature review to prove that it remains an unresolved issue.
2. Focus on making substantive, innovative contributions in one specific aspect—such as proposing a new model structure, a new training method, or a new data construction approach. Simply combining existing technologies will only result in a system implementation report rather than an original research paper.
3. The validation of experimental results should preferably be based on advanced, commonly used datasets in the research field.

---

### Official Review · Reviewer_5mWz · 2025-10-26

**Soundness:** 2
**Presentation:** 2
**Contribution:** 2
**Rating:** 4
**Confidence:** 3

**Summary:**

This paper proposes a unified data-centric + model-centric pipeline for Facial Emotion Recognition (FER). On the data side, it claims to (i) remove noisy and duplicate samples, (ii) refine faces via MediaPipe Face Mesh masks, and (iii) perform class-aware augmentation to rebalance labels. On the model side, it introduces a lightweight HyFER (MobileViT-inspired) student and Emotion-Aware Dynamic Distillation (EADD) that adaptively weights three teachers (CrossViT-18, DaViT-B, MobileViT-S) per emotion class, followed by post-distillation adversarial training (FGSM/PGD/DeepFool). Reported results: KDEF accuracy ≈ 99.6% and FER2013 accuracy ≈ 79.4% under 5-fold CV.

**Strengths:**

- Addresses both data quality and student-teacher training, with a structured, modular pipeline (Figure 1, p. 3).

- A lightweight student aimed at edge devices, with some compute stats for teachers (Table 3, p. 8).

- Per-class distillation idea. The emotion-aware weighting over multiple teachers is a reasonable intuition for FER’s class-dependent difficulty (Eq. (5), p. 6).

- Adversarial robustness is at least considered (Table 5, p. 9), which is rare in many FER papers.

**Weaknesses:**

**Limited novelty**: While the performance boost is clear and observable, when it comes to the methodology, it appears to be in engineering and pipeline design; the degree of methodological novelty seems limited.

The proposed EADD reduces to per-class soft weighting of multiple teachers by their per-class validation accuracy (Eq. 5)—a straightforward ensembling heuristic rather than a new KD principle. Closely related adaptive/multi-teacher KD ideas (attention-, meta-, or policy-weighted) already exist. On the data side, the pipeline relies on manual relabeling, cosine-similarity dedup, and face masking—standard hygiene steps that is also hard to find novel.

**Complexity**:
The system stacks three teachers + EADD + an adversarial phase, yet the student lags significantly behind its best teacher model performance in FER 2013. [Table 2,3]

**Narrow Dataset**: The proposed methodology is tested on a single model and evaluated only in two datasets. More rigorous evaluation on WILD-FER datasets like RAF-DB or AffectNet and/or testing with various student/teacher models could help see the proposed method's effect more clearly.

**Limited Details.**: The adversarial phase (FGSM/PGD/DeepFool) lacks specific details, i.e. attack budgets.

**Questions:**

1. Can you report results on the official FER2013 test split (not 5-fold CV), with mean±std over several seeds, so we can compare apples-to-apples with prior work?

2. Could you also conduct ablation within data-cleansing part? It is hard to see which method of the data cleansing contribute to the most.

3. How does your method compare to (a) a learned/adaptive multi-teacher KD baselines (please include few, if possible) and (b) label-noise-aware training (e.g., Cleanlab/FERPlus labels) without EADD?

4. Could you also test your method on more IN-THE-WILD FER data like RAF-DB or Affectnet?

---

### Official Review · Reviewer_LAF4 · 2025-10-30

**Soundness:** 1
**Presentation:** 1
**Contribution:** 2
**Rating:** 2
**Confidence:** 5

**Summary:**

The paper proposes a “unified” framework for Facial Emotion Recognition (FER) that combines:
1. A pipeline for data cleaning of the FER2013 and KDEF datasets (duplicate removal, label correction, class-balanced augmentation).
2. A hybrid CNN-Transformer model (HyFER) to perform emotion classification.
3. A dual-phase training strategy, including knowledge distillation and post-distillation adversarial training.

Experiments on FER2013 and KDEF show small performance gains compared to previous FER methods.

**Strengths:**

- **Some empirical effort**: Includes logical ablations for choice of teacher model and adversarial robustness (Tables 2-5).
- **Focus on data quality**: The motivation that FER datasets contain noise and imbalance is valid and relevant

**Weaknesses:**

### 1. **Extremely limited novelty**
- The “data-centric” pipeline consists of standard manual cleaning and simple augmentations (flip/rotate/scale) - techniques already standard in image preprocessing.
- The model (HyFER) is a MobileViT-inspired CNN+Transformer with no novel modifications beyond that extent. The choice to use a hybrid model over a uniform architecture is not discussed, and the training scheme (multi-teacher KD + adversarial fine-tuning) is entirely standard.
  The “Emotion-Aware Dynamic Distillation (EADD)” is just a weighted KD loss using validation accuracy as a softmax weight - this is a minor, mechanical variation.
- There is no genuinely novel model component or data preprocessing method.

### 2. **Weak empirical validation**
- Only two small, dated datasets (FER2013, KDEF) are used, both with low resolution and controlled conditions. These benchmarks are long saturated and not suitable for an ICLR level paper. Some models in Table 1 already achieve a highly saturated (97%+) accuracy on KDEF.
- Improvements over EA-Net in Table 1 are negligible except for a couple cases, calling into question the value of the proposed method, especially given the above weaknesses.

### 3. **Poor writing and presentation**
- There are numerous grammatical errors throughout the paper (improper capitalization (Line 1, 152), spelling mistakes (Line 331,  351), etc.)
- Many citations are misplaced or duplicated (e.g., repeated CrossViT and DaViT entries in references, two different citations used for KDEF dataset).
- The caption for Figures 1 and 2 are entirely uninformative.

**Questions:**

1. Why were only FER2013 and KDEF used? Have you evaluated on any more modern FER benchmarks (AffectNet, RAF-DB, ExpW)?
2. What are the “emotion-specific strengths” for each teacher model? How are they identified?
3. The data cleaning steps mention "manual visual inspection". How reproducible is this?

---

### Official Review · Reviewer_tZVX · 2025-10-31

**Soundness:** 1
**Presentation:** 2
**Contribution:** 1
**Rating:** 0
**Confidence:** 3

**Summary:**

This paper proposes the improve Facial Emotion Recognition (FER) by improving both data quality and model design. For data quality, the authors remove images with wrong labels and generate augmented images for classes with a small number of samples. For model training, the authors use multi-teacher knowledge distillation and adversarial training. The proposed method achieves somewhat good accuracy in the experiments.

**Strengths:**

S1: The paper is written in a relatively clear manner and easy to read.

**Weaknesses:**

Computer vision is not my expertise but I have to say that the paper looks like an experiment report from undergraduate students rather than a technical paper targeted for a top conference. My apologies if this sounds offensive.

W1: Novelty is very limited. The image quality enhancement in Section 3.1.1 relies on manual inspection, which does not scale. The data augmentation in Section 3.1.2 uses standard data augmentation methods and simply balances the number of samples from different classes. As stated by the authors in the Introduction, transfer learning and adversarial training have been used in existing works, and thus the training methods of the paper are also not novel.

W2: The main experiment results in Table 2 are weak; the proposed method slightly outperforms existing methods for some metrics but underperforms the baselines for other metrics. Normally, the best method for each metric should be marked in bold but the authors simply mark the proposed method all in bold. The experiment part also focuses entirely on model design and lacks an ablation study for the data quality part of the proposed methods.

W3: The writing of the paper can be significantly improved. There are many typos. Moreover, the related work can be more concentrated on the methods for data quality enhancement in computer vision and the model designs to improve accuracy.

**Questions:**

NA

---

### Note · Authors · 2025-11-13

I have read and agree with the venue's withdrawal policy on behalf of myself and my co-authors.